# Patient-Tailored Stenting versus Endarterectomy for the Treatment of Asymptomatic Carotid Artery Stenosis

**DOI:** 10.3390/jpm12060882

**Published:** 2022-05-27

**Authors:** Michael T. Bounajem, Ramesh Grandhi, Craig J. Kilburg, Philipp Taussky

**Affiliations:** Clinical Neurosciences Center, Department of Neurosurgery, University of Utah, Salt Lake City, UT 84132, USA; michael.bounajem@hsc.utah.edu (M.T.B.); ramesh.grandhi@hsc.utah.edu (R.G.); craig.kilburg@hsc.utah.edu (C.J.K.)

**Keywords:** carotid endarterectomy, carotid artery stenting, stroke, carotid artery stenosis, asymptomatic, stroke prevention

## Abstract

Carotid artery stenosis is a major cause of acute ischemic strokes in adults. Given the consequences and sequelae of an acute ischemic stroke, intervention while patients are still asymptomatic is a key opportunity for stroke prevention. Although carotid endarterectomy has been the gold standard of treatment for carotid stenosis for many years, recent advances in carotid stenting technology, practitioner experience, and dual antiplatelet therapy have expanded the use for treatments other than endarterectomy. Review of the current literature has demonstrated that endarterectomy and carotid artery stenting produce overall similar results for the treatment of asymptomatic carotid stenosis, but certain factors may help guide physicians and patients in choosing one treatment over the other. Age 70 years and older, renal disease, poor medication compliance, and unstable plaque features all portend better outcomes from endarterectomy, whereas age under 70 years, high cervical location of disease, cardiac disease, and reliable medication compliance favor stenting. The decision to pursue endarterectomy versus stenting is therefore complex, and although large studies have demonstrated similar outcomes, the approach to treatment of asymptomatic carotid stenosis must be optimized for each individual patient to achieve the best possible outcome.

## 1. Introduction

With an incidence of nearly 800,000 strokes per year in the United States, it is unsurprising that a tremendous amount of effort has been dedicated to studying the etiology and treatment of stroke [1]. Given that approximately 80% of strokes occur in previously asymptomatic patients, it is imperative that opportunities for pre-morbid intervention are thoroughly assessed and acted upon [2]. Asymptomatic carotid artery stenosis represents a prime opportunity for pre-stroke intervention. With a prevalence of approximately 2 million individuals, asymptomatic carotid artery stenosis is present in a significant portion of the population [1]. Additionally, it has been previously demonstrated that 15–20% of strokes are attributable to carotid artery stenosis, meaning that effective intervention in asymptomatic patients has the potential to dramatically reduce the incidence of strokes [3,4]. The question of what form intervention should take is therefore a crucial one. Although carotid endarterectomy (CEA) has previously been considered the gold standard for the treatment of carotid artery stenosis [5], recent advances in carotid artery stenting (CAS) and dual anti-platelet therapy have called into question the most optimal management of carotid artery stenosis in general, and asymptomatic carotid stenosis in particular. Although randomized control trials have demonstrated equivalence between these two treatment options, secondary analyses have shown that certain subgroups have notably different outcomes [6]. The purpose of this review is therefore to examine factors that would guide a custom-tailored approach to treatment of asymptomatic carotid artery stenosis with CEA versus CAS.

## 2. Carotid Endarterectomy

Since its original description in 1954 by Eastcott et al. [7], the CEA has become a mainstay of treatment for carotid stenosis. Its efficacy was confirmed in 1991 through the North American Symptomatic Carotid Endarterectomy Trial (NASCET) [5]. In this randomized controlled trial, patients with 70–99% carotid artery stenosis (severe stenosis) and recent transient ischemic symptoms or nondisabling stroke were randomized to best medical management (including daily aspirin therapy) with or without CEA. Patients undergoing surgery were found to have a significantly lower rate of strokes (absolute risk reduction 17 ± 3.5%, *p* < 0.001) and lower rates of major or fatal strokes (absolute risk reduction 10.6 ± 2.6%, *p* < 0.001) on the ipsilateral side of surgery [5].

Given its success in symptomatic carotid artery stenosis, investigators then asked whether CEA might provide sufficient benefit in asymptomatic patients as well. This was answered by the Asymptomatic Carotid Atherosclerosis Study (ACAS) in 1995. In this randomized controlled trial, asymptomatic patients with 60% or greater stenosis were randomized to best medical management (including daily aspirin therapy) with or without CEA [8]. With an average follow-up period of 2.7 years, patients in the surgical arm had lower rates of ipsilateral stroke or death (5.7%) compared with patients in the medical arm (11%; aggregate risk reduction of 53% with 95% confidence interval from 22% to 72%) [8]. The benefit of CEA in asymptomatic patients was again confirmed by the Asymptomatic Carotid Surgery Trial (ACST). In this randomized controlled trial, patients with carotid stenosis of 60% or greater were randomized to best medical management (including antiplatelet therapy) with or without CEA [9]. At five-year follow-up, patients in the surgical arm had a lower rate of strokes compared with those in the medical arm (6.4% versus 11.8%, respectively; net difference 5.4% with 95% CI (3.0–7.8), *p* < 0.0001) [9].

## 3. Carotid Artery Stenting

Endovascular treatment of carotid artery stenosis began in 1980, when Mathias et al. [10] proposed the use of carotid angioplasty. Although simple and intuitive, carotid angioplasty alone was beset by issues of re-stenosis and carotid artery dissections, the latter of which became the impetus for the first carotid artery stent placement [10]. Since its original description, the use of CAS has increased remarkably due to the technique’s minimally invasive nature compared with traditional CEA. Although it is now considered to be a safe and efficacious treatment of carotid stenosis, outcomes in CAS with balloon angioplasty were not always as positive for a variety of issues, one being the propagation of emboli distal to the stenotic segment. The first major technical advancement in this area came in the form of the implementation of a distal occlusive balloon by Theron et al. [11] in 1990. By inflating a balloon distal to the stenotic segment and aspirating after angioplasty, the surgeons were able to remove potential embolic particles from the circulation and minimize the risk of embolism as a result [12]. This was successful to the point that Theron et al. [13] published a series in which 43 patients underwent angioplasty alone with no embolic complications; additionally, 93 patients underwent angioplasty with stenting, and only one embolic complication was observed. Embolic protection devices have continued to evolve, and their use has become widely accepted in CAS.

Another significant reason for poorer outcomes in initial studies of CAS was the lack of sufficient experience. CEA had been performed for approximately 35 years before its first major trial documenting efficacy, whereas CAS, which was developed in the 1980s and 1990s, has continued to evolve in practitioner technique and technology concomitant with trials assessing its safety and efficacy. It has therefore been well documented that practitioner experience has a significant effect on outcomes in CAS [14,15]. For example, Ahmadi et al. [16] documented a significant decrease in the rates of stroke and mortality when comparing the initial 80 cases of CAS and the subsequent 240 patients (*p* < 0.03). At an institutional level, Wholey et al. [17] demonstrated that institutions that had performed >500 cases had a stroke and mortality rate of 1.56%, whereas institutions that had performed ≤50 cases had a stroke and death rate of 4.04%. Although it is unsurprising that practitioner experience is an important factor in outcomes after CAS, it was often overlooked in early studies and is at least in part responsible for recent improvements in CAS outcomes.

## 4. CEA versus CAS: A Direct Comparaisons

Given that CEA and CAS both provide efficacious treatment for carotid artery stenosis, it is crucial to ask whether one is superior to the other, with simultaneous minimization of morbidity and mortality. Several multicenter, randomized controlled trials have been conducted to address this exact question in asymptomatic carotid artery stenosis (Table 1). In 2010, the results of The Carotid Revascularization Endarterectomy versus Stenting Trial (CREST) were initially published. This randomized controlled trial compared rates of perioperative stroke, myocardial infarction (MI), and death in both symptomatic and asymptomatic patients with carotid stenosis randomized to CEA or CAS [18]. For asymptomatic patients to be included, they had to have stenosis of ≥60% (diagnosed on formal angiography), ≥70% (diagnosed by ultrasound), or ≥80% (when diagnosed by CT or MR angiography in the context of stenosis of 50% to 69% diagnosed on ultrasound) [18]. In the perioperative period, strokes occurred more frequently in the CAS group compared with the CEA group (4.1% versus 2.3%, *p* = 0.01); however, MIs occurred more frequently in the CEA group than in the CAS group (1.1% versus 2.3%, *p* = 0.03) [18]. It was noted that patients under the age of 70 years had better outcomes after CAS, whereas those over the age of 70 had better outcomes after CEA (*p* = 0.02) [18]. Additionally, symptomatic patients had higher risk of stroke and death after CAS, whereas asymptomatic patients had similar rates after both CAS and CEA [18].

A subsequent trial, Carotid Angioplasty and Stenting Versus Endarterectomy in Asymptomatic Subjects who are at Standard Risk for Carotid Endarterectomy with Significant Extracranial Carotid Stenotic Disease (ACT-1), was completed in 2013 to assess for stroke, MI, and death specifically in asymptomatic patients with severe carotid artery stenosis undergoing CEA or CAS. Enrolled patients had severe stenosis, with an average of 73% stenosis in both CEA and CAS groups [19]. The trial lasted for 8 years and enrolled 1453 patients from 62 institutions across the U.S [6,19]. Although it was stopped early because of slow enrollment, ACT-1 demonstrated insignificantly different rates of stroke and death at 30 days between CEA and CAS (1.7% versus 2.9%, respectively, *p* = 0.33), as well as similar 5-year stroke freedom rates of 97.4% and 93.1%, respectively (*p* = 0.44) [20,21].

The Second Asymptomatic Carotid Surgery Trial (ACST-2) was completed in 2020, with an objective of comparing CEA and CAS in asymptomatic patients with carotid stenosis of 60% or greater. In total, 3625 patients were enrolled, with 1814 undergoing CEA and 1811 undergoing CAS [22]. It demonstrated similar rates of periprocedural disabling stroke and death in CEA and CAS (1%). Additionally, the rate of nondisabling stroke was similar at 2%, with five-year stroke estimates insignificantly different at 4.5% in CEA and 5.3% in CAS (*p* = 0.33) [22].

The heretofore completed trials demonstrate safety and efficacy of both CEA and CAS in treating asymptomatic carotid artery stenosis. There are, however, major differences in implications of the two procedures, and therefore the guidelines to choose one over the other are poorly defined. Fortunately, certain subgroups have already been shown to benefit in asymmetric fashion from CEA versus CAS (Table 2). For example, in CREST, older patients had better outcomes after CEA suggesting that older patients should generally be counseled toward CEA [18,23].

Comorbid conditions may also significantly influence the decision between CEA and CAS. MI was found to occur with greater frequency in patients undergoing CEA, meaning that patients with significant cardiac history may incur greater risk for cardiac complications with CEA as compared with CAS [18]. Renal disease, which is commonly encountered in this patient population, may also influence the decision to pursue CAS, which includes a significantly greater contrast burden intra-procedurally, something that patients with significant kidney dysfunction may not tolerate. Other factors to be considered include the need for strict medication compliance postoperatively. As is the case for most stenting procedures, dual antiplatelet therapy is mandated in the immediate post-CAS period to prevent against thromboembolic complications. Patients who have a history of poor medication compliance or who have barriers to obtaining daily medications may therefore be better candidates for CEA in order to avoid the need for strict dual antiplatelet therapy.

Individual patient anatomy is also an important consideration. For example, in patients with diseased carotid segments located superiorly near the angle of the mandible, CEA is at times impractical whereas CAS is unencumbered by the surrounding structures. Another structural consideration is that of the stenosis itself. Intraplaque hemorrhage, which is now readily identifiable on noninvasive imaging [24], is itself a risk factor for acute stroke [25]. Because of its unstable characteristics, the presence of intraplaque hemorrhage increases risk for causing embolic complications during manipulation by a guidewire or balloon as it is passed through the stenotic segment. Meta-analyses have demonstrated greater rates of perioperative MI, stroke, and death in patients who undergo CAS in the presence of intraplaque hemorrhage [26].

## 5. Discussion

The treatment of carotid stenosis poses significant questions, particularly regarding the timing and modality of treatment. CEA has traditionally been the most efficacious method of treatment for carotid stenosis, but with the expanding utility of dual antiplatelet therapy and increasingly positive outcomes after CAS, this standard must be appropriately scrutinized. This issue is particularly crucial when treating cases of asymptomatic carotid stenosis, as the benefit of preventing strokes is significant but the consequences of adverse events in an asymptomatic patient are severe.

In the aforementioned randomized controlled trials, CEA and CAS have generally been found to have similar rates of stroke prevention and complications, but subgroup analyses demonstrate areas of asymmetry in outcomes [18,20,22,27]. For example, CEA was favored in patients over the age of 70 years, patients with renal disease or with intraplaque hemorrhage, and those with lesser medication compliance, whereas CAS was favored in patients with high cervical location of disease, and patients with cardiac conditions. Additionally, females were subject to a higher rate of stroke and death after CAS, favoring CEA instead [28]. In both CAS and CEA, it is crucial for the practitioner to consider his or her own experience when guiding patients through the process of choosing an intervention.

Although the extant trials have provided evidence of safety and efficacy for both CEA and CAS, further studies are still required to elucidate the nuances of indications for each. The Carotid Revascularization and Medical Management for Asymptomatic Carotid Stenosis Trial (CREST-2) is currently underway; it consists of two separate comparisons: (1) carotid endarterectomy versus best medical therapy, and (2) carotid artery stenting versus best medical therapy in patients with high-grade asymptomatic carotid stenosis [29]. Other factors being actively scrutinized include characteristics of the stenosis itself, such as the presence of intraplaque hemorrhage, plaque calcification, and plaque instability, all of which may portend a greater chance for future strokes [30]. To identify these factors, newer screening methods have been increasingly used and studied. For example, >2 microemboli per hour identified on transcranial Doppler with microemboli detection in patients with asymptomatic carotid stenosis has been associated with significantly greater risk for strokes compared with similar patients without identified microemboli [31]. Standard duplex ultrasonography has also been used to assess for plaque echolucency, which has been associated with a greater chance for strokes [32,33,34]. These factors may eventually guide decision making to an even greater extent than the presently considered characteristics such as percent stenosis when determining the need for treatment of carotid stenosis.

## 6. Conclusions

With a robust body of literature examining the safety and efficacy of CEA and CAS, it is generally well established that these two procedures are safe and beneficial for the treatment of carotid artery stenosis. Nevertheless, the patient population in need of these treatments is extremely diverse, and the decision to undertake an endovascular versus open procedure is a difficult one for both patients and practitioners. Factors that can aid in this crucial decision include patient age, vessel and plaque morphology, medication compliance, and practitioner experience. As the techniques and technological aspects of the treatment of carotid artery stenosis continue to evolve, so too will the factors that help determine the ideal treatment for each individual patient.

## Figures and Tables

**Table 1 jpm-12-00882-t001:** Comparison of outcomes among carotid endarterectomy, carotid artery stenting, and best medical therapy.

Study	*n*	Intervention	Comparison	Main Outcomes	All Outcomes (%)	Key Points
ACAS	1662	CEA	BMT	Stroke, mortality	5.1 (CEA)	Significantly lower stroke rate in surgical arm
					11 (BMT)	
ACST	3120	CEA	BMT	Stroke, mortality	2.8 (CEA)	Significantly lower stroke rate in surgical arm
					4.5 (BMT)	
CREST	2502	CAS	CEA	Stroke, MI, mortality	7.2 (CAS)	Significantly fewer strokes in CEA arm; significantly fewer MIs in CAS arm
					6.8 (CEA)	Significantly lower risk of stroke and death in asymptomatic patients
ACT-1	1665	CAS	CEA	Stroke, mortality	2.9 (CAS)	No significant difference
					1.7 (CEA)	
ACST-2	1330	CAS	CEA	Stroke, MI, mortality	3.9 (CAS)	No significant difference
					3.2 (CEA)	

MI, myocardial infarction; ACAS, Asymptomatic Carotid Atherosclerosis Study; CEA, carotid endarterectomy; BMT, best medical therapy; ACST, Asymptomatic Carotid Surgery Trial; CREST, Carotid Revascularization Endarterectomy versus Stenting Trial; CAS, carotid artery stenting; ACT-1, Carotid Angioplasty and Stenting Versus Endarterectomy in Asymptomatic Subjects who are at Standard Risk for Carotid Endarterectomy with Significant Extracranial Carotid Stenotic Disease; ACST-2, Second Asymptomatic Carotid Surgery Trial.

**Table 2 jpm-12-00882-t002:** Individual patient factors favoring CEA versus CAS.

Factor	CEA	CAS
Low medication compliance	X	
Age > 70 years	X	
Intraplaque hemorrhage	X	
Female	X	
High cervical location		X
Cardiac disease		X
Renal disease	X	
Contralateral carotid occlusion		X

CEA, carotid endarterectomy; CAS, carotid artery stenting; X denotes favor to the respective intervention.

## Data Availability

Not applicable.

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
