# Peer review of "Patient-Tailored Stenting versus Endarterectomy for the Treatment of Asymptomatic Carotid Artery Stenosis"

_jpm, 2022, doi:10.3390/jpm12060882_

Round 1
Reviewer 1 Report
The review article has been put forward at a time when there is growing importance regarding Carotid artery stenting and CEA and its related use in asymptomatic carotid stenosis. However, there are a few shortcomings and here are my suggestions.
- With the growing evidence of the usefulness of intervention for asymptomatic carotid stenosis, writers have rightly bought out the importance of intervention in asymptomatic stenosis. However, the article lags in mentioning guidelines for the detection of carotid stenosis in asymptomatic patients. It is important to bring the mode of investigation and frequency (annual or biannual) of investigation to detect an asymptomatic patient with carotid stenosis.
- The author has tried to compare the results of CEA v/s CAS in asymptomatic carotid stenosis wherein, results from the CREST trial have been mentioned. However, the CREST trial included symptomatic as well asymptomatic patients and hence substudy analysis of this trial needs to be taken into consideration which showed marginal benefit from CEA compared to CAS.
- Since both the procedure and related complications are operator dependent as described in multiple studies, with respect to asymptomatic carotid stenosis, it is important to shed light on the fact which faculty needs to be examining and further treating the patient.
- There has been no mention with regards to the use of dual antiplatelets after the procedure which needs to be highlighted, as long-term drug compliance is one of the important issues.
- The article also does not give clarity on the percentage of stenosis in the asymptomatic patients who needs to undergo the procedure. Also, there needs to be more evidence regarding the usefulness of CEA in patients over the age of 70, since such patients are more predisposed to myocardial infarction.
- Lastly, it would be helpful to bring out a schematic chart outlining which patient should undergo CEA or CAS and in which cases either of the procedure can be preferred with respect to patient preferences.
Author Response
The review article has been put forward at a time when there is growing importance regarding Carotid artery stenting and CEA and its related use in asymptomatic carotid stenosis. However, there are a few shortcomings and here are my suggestions.
With the growing evidence of the usefulness of intervention for asymptomatic carotid stenosis, writers have rightly bought out the importance of intervention in asymptomatic stenosis. However, the article lags in mentioning guidelines for the detection of carotid stenosis in asymptomatic patients. It is important to bring the mode of investigation and frequency (annual or biannual) of investigation to detect an asymptomatic patient with carotid stenosis.
RESPONSE: We are not aware of any formal recommendations for screening asymptomatic patients to identify a new diagnosis of carotid stenosis. If the reviewer can provide guidance we will note those in the manuscript.
The author has tried to compare the results of CEA v/s CAS in asymptomatic carotid stenosis wherein, results from the CREST trial have been mentioned. However, the CREST trial included symptomatic as well asymptomatic patients and hence substudy analysis of this trial needs to be taken into consideration which showed marginal benefit from CEA compared to CAS.
RESPONSE: Although it is true that CREST included both symptomatic and asymptomatic patients, it was a landmark study in the comparison of CEA and CAS, and given that the objective of this work was to compare these two interventions, we believe it would be improper to exclude it. The main difference between symptomatic and asymptomatic groups outlined in CREST was a similar risk of stroke and death between CEA and CAS in asymptomatic patients, but a higher risk after CAS in symptomatic patients. Revisions have been made to highlight this point on lines 116-119 and 124-125.
Since both the procedure and related complications are operator dependent as described in multiple studies, with respect to asymptomatic carotid stenosis, it is important to shed light on the fact which faculty needs to be examining and further treating the patient.
RESPONSE: This is discussed on lines 93-106. We have added “among practitioners” to draw attention to that point. We have also made an addition on lines 211-213 to reinforce this point.
There has been no mention with regards to the use of dual antiplatelets after the procedure which needs to be highlighted, as long-term drug compliance is one of the important issues.
RESPONSE: Postprocedural dual antiplatelet use, drug compliance, and the consequences of poor medication adherence are specifically discussed on Page 4, lines 173-179. Medication compliance as one of the factors that may favor one procedure over the other is also mentioned in Table 2, line 208, and in the abstract.
The article also does not give clarity on the percentage of stenosis in the asymptomatic patients who needs to undergo the procedure. Also, there needs to be more evidence regarding the usefulness of CEA in patients over the age of 70, since such patients are more predisposed to myocardial infarction.
RESPONSE: The percentage of stenosis is included in each RCT description. Additions have been made to pages 3 (lines 116-119) and 4 (lines 144-155) to reflect this.
Better outcomes after CEA in patients >70 was a finding from CREST and is a well-established point of guidance among practitioners that perform both procedures. We have reversed the wording on lines 162-163 to make this clearer.
Lastly, it would be helpful to bring out a schematic chart outlining which patient should undergo CEA or CAS and in which cases either of the procedure can be preferred with respect to patient preferences.
RESPONSE: Table 2 provides a comparison of factors suggestive of which patient factors would lead one to choose CEA or CAS.
Reviewer 2 Report
This manuscript is a review of treatment for asymptomatic carotid artery stenosis. The authors recommended the patient-tailored treatment. This strategy has already been reported in symptomatic patients and is not novel. Medical treatment for asymptomatic patients has also been advanced, and medical treatment should be mentioned. I feel that it is necessary to examine which lesions can be symptomatic.
Author Response
This manuscript is a review of treatment for asymptomatic carotid artery stenosis. The authors recommended the patient-tailored treatment. This strategy has already been reported in symptomatic patients and is not novel. Medical treatment for asymptomatic patients has also been advanced, and medical treatment should be mentioned. I feel that it is necessary to examine which lesions can be symptomatic.
RESPONSE: The reviewer states that patient-tailored treatment has already been reported in symptomatic patients and is therefore “not novel” for asymptomatic patients. We respectfully disagree, since the workup, management, and treatment of asymptomatic patients differs in important aspects from those of symptomatic patients. Asymptomatic patients offer a unique opportunity for stroke prevention, but with the added risk of incurring adverse events in an otherwise asymptomatic patient. The practitioner’s attitude, skill set, and dialogue with individual patients must therefore specifically cater to these issues when taking care of asymptomatic patients, something we believe is extremely important and distinct from considerations in symptomatic patients.
Page 5 paragraph 4 discusses factors that may be indicative of higher stroke risk in asymptomatic patients, as well as imaging studies that are now being used to identify these patients.
Reviewer 3 Report
The review analyzes the frequently debated topic of CEA versus CAS, aiming to provide indications to guide practitioners in the selection of the best solution for the patient. The paper is well organized, with an adeguate introduction that highlight also the historical aspects that are important to understand how the debate has developed through the years and what are the open questions. A huge amount of literature has been produced about this topic, so the originality is not the best quality of the paper, that does not add indications that were not already known.
The writers have examinated the main trials published about the comparison between CEA and CAS. Some conclusions, however, are deduced only from the CREST trial, as for example that in patients younger than 70 years CAS is a better option. The CREST trial has been critized during the years, for example by Naylor et al. (Naylor AR. “Riding on the CREST of a wave”. Eur J VascEndovascSurg 2010; 39, 523-526), and Setacci et al. (Setacci C, De Rango P A light in the shadows in the carotid stenting. Eur J VascEndovascSurg 2010; 39: 527-528). Although the literature demonstrates clearly that CEA is superior in older patients, it is not very clear if younger patients may benefit from CAS, and this indication should not be based on one single trial, considering the bias that the CREST may have, as per Naylor and Setacci.
In order to reinforce the thesis of the writers more literature should be included, for example:
- Müller MD, Lyrer P, Brown MM et al. “Carotid artery stenting versus endarterectomy for treatment of carotid artery stenosis”. Cochrane Database Syst Rev. 2020 Feb 25;2;
- Li Y, Yang JJ, Zhu SH, Xu B et al. “Long-term efficacy and safety of carotid artery stenting versus endarterectomy: A meta-analysis of randomized controlled trials”. PLoS One. 2017 Jul 14;12;
- Moresoli P, Habib B, Reynier P et al. Carotid Stenting Versus Endarterectomy for Asymptomatic Carotid Artery Stenosis: A Systematic Review and Meta-Analysis. Stroke. 2017 Aug;48(8):2150-2157;
- Sardar P, Chatterjee S, Aronow HD et al. Carotid Artery Stenting Versus Endarterectomy for Stroke Prevention: A Meta-Analysis of Clinical Trials. J Am CollCardiol. 2017 May 9;69(18):22
- Texakalidis P, Chaitidis N, Giannopoulos S et al. Carotid Revascularization in Older Adults: A Systematic Review and Meta-Analysis. World Neurosurg. 2019 Jun; 126:656-663.e1.
About the other indications, some are not well explained, for example, in the table 2 is shown that CEA is favourable for the female sex, while CAS is the best solution for patients with CCO and previous radiation neck therapy; it should be explained why, citing the sources, such as, for example, the writers have done for the intraplaque hemorrage, that is well explained.
The quality of the language is good, the paper is clear and easy to understand.
Overall, the paper is well written and well organized, but it lacks originality, because a lot of literature has been produced about the manner, and this review does not add some particular indication that was not known before. It is, anyway, clear and useful for the practitioners, in order to guide them in the selection of the best solution. Adding some sources could reinforce the hypotesis of the paper; futhermore, the indication about the age should be reconsidered, because the evidence that CEA is better in older patients is very strong, but the evidence that CAS is better in younger patients is not confirmed at all.
Author Response
The review analyzes the frequently debated topic of CEA versus CAS, aiming to provide indications to guide practitioners in the selection of the best solution for the patient. The paper is well organized, with an adeguate introduction that highlight also the historical aspects that are important to understand how the debate has developed through the years and what are the open questions. A huge amount of literature has been produced about this topic, so the originality is not the best quality of the paper, that does not add indications that were not already known.
The writers have examinated the main trials published about the comparison between CEA and CAS. Some conclusions, however, are deduced only from the CREST trial, as for example that in patients younger than 70 years CAS is a better option. The CREST trial has been critized during the years, for example by Naylor et al. (Naylor AR. “Riding on the CREST of a wave”. Eur J VascEndovascSurg 2010; 39, 523-526), and Setacci et al. (Setacci C, De Rango P A light in the shadows in the carotid stenting. Eur J VascEndovascSurg 2010; 39: 527-528). Although the literature demonstrates clearly that CEA is superior in older patients, it is not very clear if younger patients may benefit from CAS, and this indication should not be based on one single trial, considering the bias that the CREST may have, as per Naylor and Setacci.In order to reinforce the thesis of the writers more literature should be included, for example:
- Müller MD, Lyrer P, Brown MM et al. “Carotid artery stenting versus endarterectomy for treatment of carotid artery stenosis”. Cochrane Database Syst Rev. 2020 Feb 25;2;
- Li Y, Yang JJ, Zhu SH, Xu B et al. “Long-term efficacy and safety of carotid artery stenting versus endarterectomy: A meta-analysis of randomized controlled trials”. PLoS One. 2017 Jul 14;12;
- Moresoli P, Habib B, Reynier P et al. Carotid Stenting Versus Endarterectomy for Asymptomatic Carotid Artery Stenosis: A Systematic Review and Meta-Analysis. Stroke. 2017 Aug;48(8):2150-2157;
- Sardar P, Chatterjee S, Aronow HD et al. Carotid Artery Stenting Versus Endarterectomy for Stroke Prevention: A Meta-Analysis of Clinical Trials. J Am CollCardiol. 2017 May 9;69(18):2266-2276.
- Texakalidis P, Chaitidis N, Giannopoulos S et al. Carotid Revascularization in Older Adults: A Systematic Review and Meta-Analysis. World Neurosurg. 2019 Jun; 126:656-663.e1.
About the other indications, some are not well explained, for example, in the table 2 is shown that CEA is favourable for the female sex, while CAS is the best solution for patients with CCO and previous radiation neck therapy; it should be explained why, citing the sources, such as, for example, the writers have done for the intraplaque hemorrage, that is well explained.
The quality of the language is good, the paper is clear and easy to understand.
Overall, the paper is well written and well organized, but it lacks originality, because a lot of literature has been produced about the manner, and this review does not add some particular indication that was not known before. It is, anyway, clear and useful for the practitioners, in order to guide them in the selection of the best solution. Adding some sources could reinforce the hypotesis of the paper; futhermore, the indication about the age should be reconsidered, because the evidence that CEA is better in older patients is very strong, but the evidence that CAS is better in younger patients is not confirmed at all.
RESPONSE: Text regarding the superiority of CAS in younger patients has been removed. References and discussion of other factors such as female sex have been added. We appreciate the suggested references and have now incorporated several, but we wanted to stay away from secondary literature and mainly focus on the RCTs available given the available level of evidence.
Round 2
Reviewer 3 Report
The paper is well organized, with an adeguate introduction. A huge amount of literature has been produced about this topic, so the originality is not the best quality of the paper, that does not add indications that were not already known.
The writers have examinated the main trials published about the comparison between CEA and CAS. The description of the trials has been amplified and is well organized. The table 2 has been modified in a good way, the CEA procedure is for sure the better option for older patients. Anyway, in the discussion, the fact that CAS may be better for younger patients is written again; it should be better to write only that CEA is better for older patients, that doesn't mean that CAS is better for the younger ones, also because only the CREST trial found this data. It should be better to write that CREST has shown this data (as it is already written in the paper) without suggesting an indication.
The references have been improved.
The quality of the language is good, the paper is clear and easy to understand.
Overall, the paper has been improved from the last review.